# Treatment of Double-Refractory Chronic Lymphocytic Leukemia—An Unmet Clinical Need

**DOI:** 10.3390/ijms25031589

**Published:** 2024-01-27

**Authors:** Przemysław Zygmunciak, Tadeusz Robak, Bartosz Puła

**Affiliations:** 1Department of Hematology, Institute of Hematology and Transfusion Medicine, 02-776 Warsaw, Poland; zygmunciakprzemyslaw@gmail.com (P.Z.); bpula@ihit.waw.pl (B.P.); 2Department of Hematology, Medical University of Lodz, 93-510 Lodz, Poland; 3Department of General Hematology, Copernicus Memorial Hospital, 93-510 Lodz, Poland

**Keywords:** BCL-2 inhibitors, BTK inhibitors, BTK degraders, CAR-T, PI3K inhibitors, chronic lymphocytic leukemia

## Abstract

Recent years have seen significant improvement in chronic lymphocytic leukemia (CLL) management. Targeting B-cell lymphoma (BCL-2) and Bruton’s kinase (BTK) have become the main strategies to restrain CLL activity. These agents are generally well tolerated, but the discontinuation of these therapies happens due to resistance, adverse effects, and Richter’s transformation. A growing population of patients who have previously used both BTK inhibitors and BCL2 suffer from the constriction of the following regimens. This review explores the resistance mechanisms for both ibrutinib and venetoclax. Moreover, we present innovative approaches evaluated for treating double-refractory CLL.

## 1. Introduction

Chronic lymphocytic leukemia (CLL) is a B-cell neoplasm characterized by typical CD19, CD5, CD20, and CD23 immunophenotypes, as well as kappa or lambda light chain restrictions [1,2]. It occurs mostly in the elderly, with the highest prevalence in North America, Europe, and Oceania and the disease incidence being 5 cases per 100,000 inhabitants per year [3]. In the United States alone, the incidence of CLL in 2023 was estimated to surpass 18,000, which accounts for 1.0% of all new cancer cases [4]. CLL pathobiology is a matter of constant investigation, which encourages the development of new therapeutical strategies and their implementation in patient care. Accordingly, five- and ten-year relative survival of CLL patients continually ameliorates with a higher proportion of long-term survival [5]. Recently introduced targeted agents, such as Bruton’s kinase inhibitors (BTKi) and B-cell lymphoma two inhibitors (BCL2i), further shape this trend. With their broad application in the prevailing standard of care, questions of the remaining treatment options after therapy failure arise. The answers to these questions are essential, since almost 50% of CLL patients who were treated with both venetoclax and covalent BTKi ultimately relapse on both drugs [6]. This review presents the emerging problem of selecting the best treatment option for CLL refractory to BTKi and BCL2i.

## 2. Mechanism of Action of BTK Inhibitors and CLL Resistance to Ibrutinib

Bruton’s kinase (BTK) is a tyrosine kinase essential for numerous B-cell processes such as differentiation, proliferation, and survival [7]. Its central role in B-cell biology underlines the fact that BTK is an intersection of B-cell receptor (BCR), Toll-like receptor (TLR), and chemokine receptor signaling, which not only assures the aforementioned actions but also encourages microenvironmental influence, adhesion, and migration [8]. Increased BTK activation is also found in several B-cell malignancies besides the physiological processes. For instance, increased BTK signaling is a precondition of leukemogenesis in IgH.ETμ mice and BTK deficiency abolishes CLL malignant transformation [9]. Moreover, the knock-down of BTK via small interfering RNAs (siRNAs) translates into decreased survival of CLL tumor cells in vitro [10]. Therefore, BTK blocking was chosen as a valid target for drug development.

The first agent from the new BTKi family was ibrutinib, which binds covalently with cysteine-481 of BTK, irreversibly inhibiting its autophosphorylation and consequently the phosphorylation of the downstream kinases [11,12]. This action translates into decreased proliferation and survival of CLL cells in vitro, as well as clinical responses in canine non-Hodgkin lymphoma (NHL) models [11,12]. Additionally, BTK is a regulator of the migration of B-cells mediated by chemokine adhesion [13]. Ibrutinib interferes with the CXCL12-CXCR4 and CXCL13-CXCR5 pathways, which might be responsible for the down-regulated tissue homing of malignant cells and lymphocytosis seen in the preliminary phase of the treatment [14]. Taking these actions into consideration, combined with its favorable safety profile, ibrutinib was soon tested in clinical trials, which led to its approval by the Food and Drug Administration (FDA) in 2013 (and by the European Medicines Agency (EMA) in 2014) for the treatment of mantle cell lymphoma (MCL) [15].

Since then, the FDA has approved three other agents: acalabrutinib in 2017, zanubrutinib in 2019, and pirtobrutinib in 2023. Additionally, orelabrutinib has been granted a breakthrough therapy designation by the FDA for the treatment of relapsed/refractory mantle cell lymphoma (MCL) in 2021. Acalabrutinib, zanubrutinib, orelabrutinib, spebrutinib, and tirabrutinib are next-generation covalent, irreversible BTKis that are characterized by increased BTK selectivity, which correlates with less pronounced adverse effects such as atrial fibrillation and hypertension [15,16,17,18]. The results from phase 2 of a study on the use of orelabrutinib in relapsed/refractory CLL were recently published and the drug achieved great efficacy, with an ORR of 92.5% and complete response (CR) seen in 21.3% of the participants, which might speed up the drug approval process for this indication [18]. Other covalent agents (spebrutinib and tirabrutinib) are currently being evaluated in CLL patients relapsing on other treatments. The results from the second phases of their clinical trials are expected shortly [19,20]. Pirtobrutinib is the first non-covalent, reversible BTKi approved recently in the US to treat adult patients with relapsed or refractory MCL after at least two lines of systemic therapy, including a covalent BTKi. The characteristics of the BTKis are displayed in Table 1.

As a drug for CLL, ibrutinib was first tested in a relapsed/refractory setting, where compared to ofatumumab, it showed better efficacy in terms of overall survival (OS), progression-free survival (PFS), and overall response rate (ORR) [32]. Intriguingly, PFS continued to increase, reaching 91% as the study period lengthened [33]. The RESONATE-2 trial further broadened ibrutinib’s indication in CLL by showing increased PFS, OS, and ORR of previously untreated CLL patients compared to chlorambucil [34,35,36].

Regardless of ibrutinib’s efficacy in clinical trials, its use may be reduced due to the resistance of CLL cells (Figure 1). This process is rather elusive since no simple mutation drives CLL tumorigenesis. However, in around 80% of ibrutinib-resistant patients, the mutations of *BTK* or Phospholipase Cγ2 (*PLCG2*), the downstream of BCR signaling, are found [37,38]. The most common BTK mutation found in ibrutinib-resistant patients is cysteine-to-serine mutation (C481S), which inhibits covalent bond formation between ibrutinib and kinase, imposing its reversible action [39]. Combined with the rather short half-life of ibrutinib, it leads to impaired inhibition of BTK and resistance [39]. *PLCG2* mutations are responsible for autonomous BCR signaling activation, and thus progression, whereas BTK remains inhibited [39,40]. Despite them being the most common mutations seen in ibrutinib-resistant patients, other mutations, including point mutations (e.g., BIRC3, CARD11, MGA, NFKBIE, RIPK1, RPS15, SF3B1, TP53, and XPO1) and chromosomal abnormalities (del8p, del18p, MYC gain/amplification, gain of 2p), may be of importance in rendering resistance to ibrutinib [41,42,43,44,45]. Some studies postulate the impact of the coexistence of BTK/PLCG2 and other mutations leading toward ibrutinib resistance. A recent study showed that mutated *EGR2* was often associated with mutated BTK, possibly cooperating in disrupting BCR signaling [46]. On the other hand, altered *BIRC3* and *NFKBIE* genes exist with wild-type BTK [46]. Of note is that the study detected a less pronounced percentage of BTK/PLCG2 mutations (65%) in ibrutinib-relapsing patients [46].

## 3. Mechanism of Action of BCL2 Inhibitors and CLL Resistance to Venetoclax

As mentioned before, dysregulated BCR signaling is one of the hallmarks of leukemogenesis. However, CLL cells develop additional mechanisms to ensure their survival. One of them, crucial for malignant cell longevity, is the evasion of apoptosis. In line with this, increased levels of anti-apoptotic proteins from the BCL2 family, including BCL2, and myeloid cell leukemia-1 (MCL1), were detected in specimens from previously untreated CLL patients [47]. Their main mechanism in preventing apoptosis lies in the fastening of the activators, such as BH-3-interacting domain death agonist (BID) and Bcl-2 Interacting Mediator of cell death (BIM), before they tie up to BCL2-associated X protein (BAX) or BCL2-antagonist/killer (BAK), which determine BH-3-dependent apoptosis [48,49]. It is essential to note that a cell using the mechanism above to ensure its survival depends solely on the continuity of the increased anti-apoptotic vs. pro-apoptotic stimulus ratio. Thus, the increase in pro-apoptotic or decreased anti-apoptotic signaling leads to the activation of BAX and BAK, triggering programmed cell death. The latter method was used to create BCL2 inhibitors.

Venetoclax was the first BCL2i accepted for the treatment of CLL. It showed promise in a relapsed CLL setting, where 89% of patients had unfavorable clinical or genetic features (e.g., del17p, unmutated *IGHV*, resistance to fludarabine), achieving an ORR of 79% and CR in 20% of the patients, among whom undetectable minimal residual disease (MDR) was confirmed in 35% [50]. Further studies confirmed the excellent outcomes of venetoclax monotherapy among CLL patients harboring deletion 17p (ORR 79%) [50]. Thus, venetoclax was soon approved by the FDA as a treatment option for CLL patients harboring deletion 17p and by the EMA for CLL patients with del17p or *TP53* mutations, who cannot use BTKi due to relapse or unsuitability and patients who do not present with the mutations and fail both chemoimmunotherapy and BTKi. Since then, venetoclax has started to gain recognition as a potent agent in CLL therapy.

Although remarkably effective, initial venetoclax therapy success may be interrupted due to accelerating resistance. Thus far, a few resistance mechanisms have been described in the literature (Figure 1).

Similar to the previously described ibrutinib resistance mechanism, the CLL cells may acquire mutations reducing the affinity of molecular drugs. These mutations (BCL2 G101V, D103Y, F104L) impair venetoclax binding, thus leading to disease relapse [51,52,53], although BCL2 point mutations are not necessary for the development of resistance. Another study conducted whole-exome sequencing of the samples acquired from CLL patients relapsing on venetoclax and found no such mutations [54]. Instead, recurrent mutations appeared in TP53, NOTCH1, CDKN2A/B, BRAF, CD274, SF3B1, and BTG1 genes, underlining the heterogeneity of the mechanisms rendering resistance [54].

Another way in which lymphoid cells may become venetoclax-resistant is through the up-regulation of other anti-apoptotic BCL-2 family members: B-cell lymphoma-extra large (BCL-XL) and MCL1 [55]. Their inhibition in vitro increases venetoclax sensitivity [55,56,57]. Overexpression of these genes may be the effect of the amplification or increased signaling from Toll-like receptor 7 (TLR7), and ergo microenvironmental stimuli such as interleukin-10 (IL10), cluster of differentiation 40 ligand (CD40L), and cytosine guanine dinucleotide-oligodeoxynucleotides (CpG-ODNs) [56,57]. Intriguingly, some studies suggest that BCL-XL may be more potent at rendering resistance since BIM preferred interacting with BCL-XL rather than MCL1 in venetoclax-treated CLL cells [58]. Notably, apart from the up-regulated anti-apoptotic BCL2 family members, the down-regulation of their pro-apoptotic counterparts occurs [56,59].

Lastly, venetoclax-resistant cells have dysregulated metabolism, including increased oxidative phosphorylation [56]. BCL2 inhibition increases cytochrome c release, interfering with mitochondrial energy production [56]. The disturbance in adenosine-3-phosphate (ATP) generation leads to AMP-activated protein kinase (AMPK) up-regulation, which guards mitochondrial homeostasis [60]. In venetoclax-resistant cells, the amplification of the 1q23 region, including AMPK, was shown [56]. As a result, the increased mitochondrial metabolism was preserved, thus maintaining cellular resistance [56]. Importantly, inhibiting both AMPK and oxidative phosphorylation leads to increased venetoclax sensitivity, which might be translated to clinical use [56].

## 4. BTK Inhibitor and BCL2 Inhibitor Combination

Since both ibrutinib and venetoclax inhibit different signaling pathways, their combination soon became of interest as a possible effective treatment option. Indeed, preclinical data have shown the synergy of both drugs in vitro, ex vivo, and in murine models, which gave rationale for the introduction of the combination in clinical trials [61,62,63]. The mechanism behind this cooperation is not yet fully understood, but the increase in dependence on BCL2 signaling in CLL cells by ibrutinib is proposed [61,62,63]. In some studies, the down-regulation of MCL1 by BTKi was observed [61,62]. In line with that, the resistance to venetoclax in CLL cells in vitro, mediated by the up-regulation of MCL1, may be interrupted by the inhibition of the BCR signaling [64]. However, the increase in BIM levels as the driving force of the synergy may be a more possible hypothesis since not every CLL sample treated with both drugs ex vivo presented with altered anti-apoptotic BCL2 protein levels [62]. Comparable outcomes were observed in the murine model [63].

Another mechanism rendering the co-efficiency of the drugs is the lymphocytosis-inducing effect of ibrutinib. Several studies describe CLL mobilization as an effect of ibrutinib’s treatment [14,65]. The microenvironment of lymph nodes protects CLL cells and decreases their vulnerability to numerous therapies, including venetoclax [66,67]. One of the proposed models of the increased resistance of CLL cells inside lymph nodes is the increased expression of MCL1, BCL-XL, and BCL2A1, being a consequence of augmented BCR, CD40, and TLR9 signaling [67,68,69,70]. Ibrutinib effectively inhibits these pathways and interrupts the resistance to venetoclax [61,68]. In the lymph nodes, ibrutinib not only decreases the pro-survival stimuli but also, as noted above, decreases the activity of CXCL12-CXCR4 signaling, leading to an efflux of CLL cells into the blood [14]. In agreement, the administration of ibrutinib leads to an average absolute lymphocyte count (ALC) increase of 66% [65]. Venetoclax targets mainly bone marrow and blood populations of CLL cells [71]. Thus, the increased mobilization of the nodal population into the bloodstream enhances the number of malignant cells affected by the pro-apoptotic venetoclax mechanism of action.

The optimistic results from preclinical studies align with already published clinical data. In a study conducted on a previously untreated but high-risk cohort, the combination of ibrutinib and venetoclax was assessed to be a valid treatment option for CLL [72]. The study was designed in three phases: the first three cycles consisted of ibrutinib alone to alleviate the potential toxicities of the combined therapy, the second phase being co-therapy with weekly dose escalation of venetoclax until 400 mg/d for 24 cycles, and the third phase in which ibrutinib could be continued in patients remaining positive for MRD [72]. Initially, the CRs (including CR with incomplete hematologic recovery (CRi)) were rare; however, the initiation of the venetoclax treatment resulted in 96% of the patients achieving CR/CRi after 18 cycles of the combined therapy with undetectable MDR (uMDR) in bone marrow in 69% of the patients, giving the rationale for using ibrutinib–venetoclax therapy as a first-line treatment of CLL [72].

The potential of this combination in naïve CLL patients has also been shown in the CAPTIVATE study, where the patients received ibrutinib for three cycles, later followed up by the co-administration of ibrutinib and venetoclax [73]. After these phases, ORR was achieved in 97% of patients with CR/CRi in 46% of the patients [73]. The study was designed to evaluate the MRD after the preliminary phase and base the following treatment on the MRD results [73]. Thus, four groups were created: patients who achieved uMRD were randomized into ibrutinib vs. placebo, and patients who did not reach uMRD criteria were randomized into the continuation of ibrutinib and venetoclax or monotherapy with ibrutinib [73]. Primary results were positive for the fixed-duration regimen as the valid first-line treatment option [73,74]. Of note is the fact that an additional advantage of the combination is the fast restoration of the hematologic milieu with the decrease in immunosuppressive T-cells and increase within antitumoral myeloid cells, which might be responsible for the reduced infection rates seen in the treated patients over time [75].

The benefits of the combination are also being investigated in relapsed/refractory settings [76,77]. In a study performed on the previously treated group (the median number of prior therapies = 1), the ibrutinib monotherapy was started for 8 weeks, which was followed by co-therapy with ibrutinib and venetoclax for 12 months [76]. Overall response was seen in 89% of the patients, with 51% achieving CR/CRi [76]. Negative MRD was seen in 53% and 36% of blood and bone marrow samples, respectively, giving promise for fixed-duration therapy in this group of patients [76]. Similar conclusions were drawn from another study concerning relapsed/refractory CLL, where the researchers found that the interruption of the ibrutinib and venetoclax therapy guided by the MRD might be a beneficial approach for this particular group of patients [78]. Considering that both BTKi and BLC2i are often used in monotherapy, and the synergy of the two drugs combined was proven, the question of the efficacy of the co-therapy among the patients who failed both treatments arose. Two retrospective studies evaluated this approach [79,80].

In the first, 13 patients who underwent both BTKi and venetoclax therapies before the administration of both drugs concomitantly were identified [79]. This group was high-risk, with 100% of the patients being *IGHV*-unmutated, two-thirds having BTK or PLCG2 mutations, 53% of the patients having complex karyotype, and almost a third having del17p [79]. The median therapy count before the treatment was six [79]. The treatment consisted of venetoclax and ibrutinib in all patients but one, who received acalabrutinib instead of ibrutinib [79]. Nine patients achieved partial response (PR), two had stable disease, and two progressed on the treatment [79]. Undetectable MRD was achieved in bone marrow or blood in three out of six tested patients [79]. The 1-year OS and PFS were 70% and 56.4%, respectively [79]. The analysis results were encouraging and gave rationale for using the combination of venetoclax and BTKi in high-risk patients to achieve disease control [79]. The second study investigated the combined therapy in both venetoclax-naïve and double-refractory patients [80]. Also, the group was high-risk, with two-thirds having del17p or *TP53* mutation, 94% of the patients having unmutated IGHV, and 88% having complex karyotype [80]. The ORR in the double-refractory patients was 100%, with 55% achieving CR [80]. The median time to the next treatment was 11.2 months; however, one patient did not need any following treatment at 17.4 months. The median OS was 27 months [80]. Importantly, both studies agree that the combined therapy with venetoclax and BTKi leads to disease control in a double-refractory setting and might be a useful bridging tactic. Of note, the BTKis used in both studies were covalent BTKis (mostly ibrutinib), and the non-covalent agents were not administered. As noted above, preclinical models show the superiority of nemtabrutinib to ibrutinib when combined with venetoclax; thus, this regimen might be beneficial and needs further evaluation [81].

## 5. Regimens after Ibrutinib and Venetoclax Failure

The number of patients previously treated with both ibrutinib and venetoclax is continuously rising and so is the problem of the optimal treatment in a double-refractory setting. In this chapter, the feasible treatment options, including non-covalent BTKis, cellular therapy, stem cell transplant, and others, will be discussed (Table 2).

### 5.1. Non-covalent BTK Inhibitors

The introduction of ibrutinib significantly changed the CLL therapeutic environment. Nonetheless, the RESONATE study’s final analysis reveals that treatment termination because of adverse events or progressive disease was prevalent, with more than half of the cohort doing so [96]. The decrease in the former was achieved by increased selectivity. However, so-called second-generation covalent BTKis, including acalabrutinib and zanubrutinib, have the same binding spot as ibrutinib; they do not circumvent the usual resistance mechanisms [97]. In response to the growing need for a treatment retaining efficacy after ibrutinib treatment relapse, new non-covalent BTKis were developed.

The first one to be granted an authorization for the treatment of adult patients with CLL or small lymphocytic lymphoma (SLL) who have received at least two lines of therapy, including a BTK and a BCL-2 inhibitor, was pirtobrutinib. This agent is a highly selective non-covalent BTKi potent in BTK C481S and wild-type [97,98]. Recently, the results regarding specifically CLL/SLL from the first-in-human phase 1/2 study testing pirtobrutinib’s effectiveness and safety profile in relapsed/refractory B-cell tumors were published [27]. The paper analyzed the responses among patients who were previously treated with BTKi [27]. The cohort had a median of three prior therapies and over 40% of patients received BCL2i as a previous agent [27]. The ORR in the group was 73.3%. Importantly, the ORR among the patients treated before with both BTKi and BCL2i was as high as 70%, which shows a great efficacy of pirtobrutinib in this group of patients [27]. However, only four patients (1.6%) achieved CR and none of them were previously treated with BCL2i [27]. Median progression-free survival (mPFS) in the group was 19.6 months [27]. The article also included a safety analysis regarding all CLL/SLL patients [27]. The most common adverse events of grade ≥3 included infections, neutropenia, and anemia [27].

Another group member is nemtabrutinib, which inhibits BTK and targets other kinases down the BCR signaling pathway, thus remaining sufficient in patients harboring C481S and PLCG2 mutations [99]. The preliminary data from phase 1/2 determined the optimal dose of 65 mg daily, given to the heavily pretreated patients (median number of prior therapies in CLL/SLL group = 4) suffering from B-cell malignancies [28]. Of note, the BTKis were formerly administered in 84% of the CLL/SLL cohort, and the C481S mutation was found in 63% of the group [28]. Almost fifty-eight percent of the patients responded to the treatment, including one complete remission [28]. Importantly, the adverse events led to the discontinuation of the therapy only in 8% of the patients [28]. Moreover, the combinations of venetoclax, nemtabrutinib, and ibrutinib were tested in vivo and the results have shown the superiority of nemtabrutinib and venetoclax over ibrutinib and venetoclax therapy, which gives the rationale for the additional studies of this combination in clinical settings [81].

Two other non-covalent BTKis went into clinical trials: vecabrutinib and fenebrutinib [29,30]. The study of the former did not proceed to phase 2 because of the lack of clinical activity of the agent [29]. The fenebrutinib study was also terminated; however, it is noteworthy that the drug-induced remission in one of the six enrolled patients harbored a C481S mutation [30].

To conclude, the non-covalent BTKis, especially pirtobrutinib, are a good choice in double-refractory CLL patients. Nemtabrutinib might be another option; however, more precise data are needed. It is noteworthy that patients who relapse during non-covalent BTKi treatment may harbor secondary BTK or PLCG2 mutations [25,26]. This mechanism of genetic escape was observed in seven patients who developed new mutations in the BTK kinase domain outside C481 (V416L, A428D M437R, L528W, T474I) [26]. Two patients remained positive for PLCG2 mutations [26]. Another study evaluated the mutations in patients relapsing on pirtobrutinib and found the following BTK mutations: T474I, T474L, M477I, and L528W, eligible for resistance [25]. Nevertheless, the approval of pirtobrutinib among double-refractory CLL patients represents a huge leap towards addressing the need for efficient therapy in this clinical group.

### 5.2. Other BCL2 Inhibitors and MCL-1 Inhibitors

Additionally, venetoclax might be one of many available BCL2i members soon. There are several new BCL2is in clinical development: sonrotoclax (BGB-11417), lisaftoclax (APG-2575), and LOXO-338. In the preclinical models, sonrotoclax achieved higher anti-tumor activity and selectivity to BCL2 than venetoclax [100]. Preliminary clinical data suggest a good safety profile and efficacy with PR or better, seen in two-thirds of the cohort treated with monotherapy, and PR with lymphocytosis or better, achieved in 72,7% of the patients receiving combination treatment with zanubrutinib [101]. At 24 weeks, three out of four patients evaluated for MRD achieved uMRD after sonrotoclax [101]. In addition, ORR was seen in 95% of the relapsed/refractory CLL patients treated with the combination of zanubrutinib and sonrotoclax, with CR achieved in 30%, PR seen in 65%, and stable disease achieved in 5% [82]. To date, four studies are evaluating the use of sonrotoclax in CLL: NCT05479994, NCT04277637, NCT06073821, and NCT04883957. Hopefully, the new data will be available soon. The initial data from phase 2 of a global study evaluating the second agent, called lisaftoclax, were recently published [83]. The cohort consisted of 141 patients, of which 17 progressed on BTKi and/or venetoclax [83]. Lisaftoclax was investigated as a single agent or combined with acalabrutinib or rituximab [83]. ORR was the highest in the lisaftoclax and acalabrutinib group (98%), followed by lisaftoclax and rituximab (87%) and monotherapy (65%) [83]. The preliminary results for the last agent, LOXO-338, concerned patients with several hematologic malignancies who relapsed on at least two prior therapy lines [84]. Of note, 68% of the patients were given BTKi before LOXO-338; however, no patient received BCL2i [84]. Initial data for CLL/SLL treatment are encouraging [84].

The induction of MCL-1, seen in venetoclax-relapsed patients, gives the rationale for blocking MCL-1 as one of the feasible tactics to overcome resistance. However, MCL-1 inhibitors are not currently investigated in this setting. One of the direct MCL-1 inhibitors, AZD5991, was about to be studied in CLL patients, but an unfavorable safety profile and low ORR led to the study’s termination [102].

### 5.3. Phosphoinositide 3-Kinase Inhibitors (PI3Kis)

BCR signaling is not solely BTK-dependent, and so the advancement of the molecular agents was not developed to decrease its activity [103,104]. Other drugs invented to inhibit the downstream route of the BCR include PI3Kis, among which three were approved for treating CLL: idelalisib, duvelisib, and recently umbralisib [104]. In relapsed/refractory CLL, the ORR and median PFS were 72% and 15.8 months and 74% and 13.3 months for idelalisib and duvelisib, respectively [85,86]. However, no patient received a prior treatment with BTKi or venetoclax [85,86]. The administration of these drugs was evaluated in a double-refractory setting where an ORR of 46.9% (CR in 5.9%) was achieved [105]. This approach led to the median PFS of only 5 months, and 78% of the patients discontinued therapy: 45.5% due to progression, 19.5% due to adverse events, and the rest because of transformation. Indeed, PI3K inhibition leads to severe adverse effects (e.g., hepatotoxicity, diarrhea and colitis, skin changes, and infections), leads often to treatment discontinuation [85,86,104]. However, the last agent—umbralisib shows a greater selectivity towards δ isotypes of PI3K, which translates to less pronounced toxicity and better tolerability [104]. Of note, umbralisib was also evaluated among patients who discontinued BTKi or PI3Ki, and in this setting, ORR was 44% and median PFS reached 23.5 months with manageable adverse events in most patients [87]. Several other agents targeting PI3K (e.g., copanlisib, zandelisib, parsaclisib) are presently being developed and tested in CLL patients [104]. To conclude, PI3Kis might be a possible treatment option; however, the old agents do not present a huge improvement in PFS in a double-refractory setting. Umbralisib and new drugs should be tested in this patient group to elicit their utility, possibly in combination with other targeted therapeutics.

### 5.4. BTK Degraders

As noted before, the relapse in ibrutinib therapy is often associated with C481S mutation, and non-covalent BTKis are a leading treatment option in this setting since they bind to BTK outside of C481. Nevertheless, a new approach—proteolysis-targeting chimeras (PROTACs)—has been carefully investigated. Their distinctive mechanism of action is necessary in a growing group of patients who are refractory to covalent and non-covalent BTKis.

PROTACs are small molecules that bind selectively to the selected protein and E3 ligase [106,107]. This action leads to the ubiquitination of said protein, followed by its proteolysis [106,107]. PROTACs were studied in preclinical models, where they significantly reduced BTK signaling, and in ibrutinib-resistant and C481S-mutated CLL cells [106,107]. Their efficacy gave the rationale for implementing this approach in a clinical setting [88,89,90]. The preliminary results are available for phase Ia/Ib of PROTAC: NX-2127 [90]. In the Ia phase of the trial, one dose-limiting toxicity was noted at 300 mg and the running dosage of 100 mg was chosen [90]. The study enrolled 17 CLL/SLL patients with a median of six prior therapies, among which 76.5% were double-refractory [90]. Additionally, fourteen out of seventeen had at least one mutation in BTK or BCL2 (C481 in 29%) [90]. The mean BTK degradation was assessed at 83% and an ORR of 33% was noted [90]. Importantly, the responses among patients resistant to BCL2i, BTKi, and non-covalent BTKi were seen [90]. Clinical trials for two other agents, BGB-16673 and NX-5948, are currently ongoing [88,89]. Of note, the cohort expansion phase of the NX-5948 trial will include CLL/SLL patients who relapsed on both BTKi and BCL2i [89].

To sum up, BTK degraders are a tempting new approach to inhibit BTK function. The initial results are promising. The results from patients who have previously received non-covalent BTKis are particularly exciting, as this patient population is expected to grow significantly in the near future.

### 5.5. Bispecific T-Cell Engagers

Therapy with bispecific T-cell engagers (BiTEs) is of interest for double-refractory CLL patients [108]. Several clinical trials are being held in this particular group (NCT04623541, NCT04923048, NCT02500407, NCT02924402, NCT04806035). This approach could even have a broader application than CAR-T therapy; however, no bispecific antibodies have been approved for CLL just yet [109,110].

Blinatumomab, a bispecific monoclonal antibody targeting CD19, has some activity in refractory Richter syndrome as a bridge to allogeneic hematopoietic stem cell transplantation (alloHCT) [111]. Moreover, two ongoing clinical trials are evaluating the combination of blinatumomab with lenalidomide (NCT02568553). In another study, blinatumomab-expanded T cells are being investigated in patients with high-risk CLL and other NHL (NCT03823365).

Epcoritamab (GEN3013; DuoBody^®^-CD3Å~CD20) is another bispecific antibody potentially useful in double-refractory CLL [91]. It is a CD20 × CD3 IgG1 T-cell engager (TCE). The preliminary results have shown a favorable safety profile and activity even in the high-risk patients (86% had del17p or *TP53* mutations, the median number of six previous therapies) [91,112]. In a phase 1b/2 trial, epcoritamab showed clinical activity in patients with high-risk CLL previously treated with two or more lines of systemic therapy, including BTKi (EPCORE CLL-1; NCT04623541). Intriguingly, the preclinical data revealed the synergy of epcoritamab and venetoclax or BTKis, giving the grounds for further investigating these combinations [113,114].

Atezolizumab is a checkpoint inhibitor that binds to the programmed-cell death ligand 1 (PD-L1). It is being investigated in a phase 1/1b study in patients with R/R B-NHL or CLL (NCT02500407).

Another BiTE currently investigated in relapsed/refractory CLL is GB261, which also has an affinity to CD20 and CD3 [115]. This antibody was created to bind weakly with CD3 with preserved strong CD20 adhesion, translating to high levels of T-cell activation [115]. Its safety and efficacy are presently investigated in relapsed/refractory B-cell non-Hodgkin lymphoma and CLL (NCT 04923048).

Another agent being explored in the same setting is mosunetuzumab (Lunsumio)—a fully humanized bispecific monoclonal antibody targeting both CD20 and CD3 [112,116]. It is being tested as a monotherapy or combined with atezolizumab (NCT02500407, NCT05091424). However, in the preclinical investigation, mosunetuzumab did activate T-cells more effectively in healthy donor samples when compared to the material from CLL patients; therefore, the additional agent may be necessary for efficient B-cell elimination [117].

Plamotamab is another BiTE studied for the treatment of refractory CLL patients (with a median number of 4.5 prior therapies), and in this group, only one patient out of five achieved CR. Still, the study is ongoing (NCT02924402) [92].

Lastly, the BiTE targeting CD47 and CD19, TG-1801, is currently being investigated for several hematological malignancies (including CLL) as a monotherapy or combined with ublituximab (anti-CD20 antibody); however, to this date, there are no data available (NCT04806035) [118].

### 5.6. Chimeric Antigen Receptor-Positive T (CAR-T) and NK (CAR-NK) Cell Therapy

Since T-cell response is impaired in CLL patients, the use of CAR-T-cell therapy was proposed as a potential treatment option, especially in heavily pretreated patients [119]. Thus, the CAR-T-cells were tested in a relapsed/refractory setting, and the overall response was promising with manageable adverse events [120,121]. Particularly durable responses were achieved in the subgroups of patients in CR [120,121]. With the expansion of the new molecular therapies, the main objective of the research in the field was the efficacy of CAR-T-cells after ibrutinib and/or venetoclax relapse [93,94,105,122,123]. Also, in this context, the results were encouraging.

Turtle et al. researched the safety and efficacy of CAR-T therapy in patients who received prior ibrutinib regimens [93]. Six of the twenty-four patients were venetoclax-refractory [93]. Additionally, the complex karyotype and/or del17p was seen in the majority of the patients [93]. ORR was high, with 71% of the patients responding to the treatment, and after restaging, it rose to 74%, with 21% of the patients achieving CR [93]. No bone marrow disease was detected in 88% of the patients who had it before the CAR-T-cell therapy; however, nodal disease eradication was less commonly noted [93]. The major adverse events correlated with the therapy were not common, although one patient died due to cytokine release syndrome (CRS) and neurotoxicity [93].

Recently, the Phase 1/2 TRANSCEND CLL 004 study results were published, in which the researchers evaluated the rationale for the use of lisocabtagene maraleucel (liso-cel) in the relapsed/refractory CLL/SLL [94]. There were 137 patients eligible for the study, and 117 received the therapy. Fifty-nine percent of the cohort experienced relapse on both BTKi and BTL2i [94]. The primary analysis target was met with 18% of the cohort achieving CR/CRi in patients who were double-refractory when administered 100 × 10⁶ CAR-T cells [94]. Additionally, 63% and 59% of these participants achieved uMRD in blood and bone marrow [94]. The study ended fatally for five participants due to adverse events, but only one of them was related to the liso-cel infusion [94].

Further evidence for the efficacy of the CAR-T-cell therapy comes from a study evaluating several therapies following venetoclax failure (mostly in relapsed/refractory CLL) [105]. The study confronted the ORR and PFS between groups treated with BTKi, PI3Ki, and cellular therapies [105]. The CAR-T-cell therapy group included 18 participants with a median of four prior therapies, and all of them were previously exposed to both BTKi and venetoclax [105]. ORR was high at 66.6%, including 33.3% of CR [105]. The median PFS in this group was nine months [105].

In conclusion, cellular therapy with CAR-T cells seems to be a promising approach in the double-refractory setting; however, adverse events such as CRS and neurotoxicity need to be taken into consideration before the use of this treatment.

Another therapy evaluated in relapsed or refractory patients suffering from CLL was anti-CD19 chimeric antigen receptor natural killer cells (CAR-NK) [95]. This approach might be superior to CAR-T therapy because of the less pronounced severity of complications [95]. In the study, five CLL patients were treated, among which three had a relapse on ibrutinib, and two were double-refractory [95]. All patients had high-risk diseases and a median number of five prior therapies [95]. After the administration of CAR-NK, three participants achieved CR, and one achieved remission of Richter’s transformation with ongoing CLL [95]. Overall, this approach was successful, with most patients accomplishing CR and a manageable safety profile [95].

### 5.7. Allogenic Hematopoietic Stem Cell Transplantation

AlloHCT has been declining in recent years due to the introduction of novel treatment options; however, it might be considered in double-refractory fit patients [124]. In the aforementioned article by Mato et al., a group of participants received alloHCT after venetoclax failure [105]. This approach has also shown efficacy in relapsed CLL patients; however, the heterogeneity of the cohort and its small size made it impossible to draw other conclusions from the study [105].

In another study, the dependency between prior therapies and the efficacy of alloHCT was investigated [125]. The study included 65 participants who were given alloHCT after at least one prior therapy with ibrutinib, venetoclax, or PI3Ki [125]. In this group, there were 17 patients previously exposed to both ibrutinib and venetoclax [125]. After 2 years, the estimated OS was 81% and PFS was 63% [125]. PFS and OS were comparable in patients receiving one vs. two novel agents and patients receiving novel agents alone vs. patients previously treated with chemoimmunotherapy [125]. Moreover, the study evaluated the differences in PFS and OS depending on the therapy directly before the alloHCT in the group exposed to both venetoclax and ibrutinib [125]. Also, no significant differences were noted, although the venetoclax–prior-alloHCT group had a much lower 12-month relapse incidence than the one observed in ibrutinib–prior-alloHCT participants (20% vs. 9.3%) [125]. Retrospective analysis of the 108 patients who received alloHCT and were priorly exposed to targeted agents has shown the great efficacy of the treatment in heavily pretreated (the median number of the prior therapies = 4), high-risk cohort with the 3-year OS of 87% and 3-year PFS of 69% [126]. Intriguingly, the 3-year OS and PFS were lower in the group who received chemoimmunotherapy solely before the transplant and were 69% and 58%, respectively [126]. However, this improvement is not solely the effect of the targeted agents but the combined result of the former and the better alloHCT management [126]. In the analyzed group, seven patients received both venetoclax and ibrutinib before the transplantation, but the specific data for this patients’ group are lacking [126].

It is noteworthy that these studies included patients who often were exposed to the targeted agents not necessarily being refractory. To conclude, alloHCT remains a well-tolerated and durable treatment option, especially in high-risk CLL patients. The eligibility for alloHCT needs to be carefully evaluated since the patient’s fitness is still a constricting factor. However, the number of alloHCT procedures performed in recent years has dramatically decreased due to the introduction of novel targeted drugs, especially BTK inhibitors and venetoclax.

## 6. Conclusions

The landscape of the CLL treatment was forever changed after the introduction of ibrutinib and venetoclax. Both agents have significantly improved patient care; however, the relapses are often, and the further therapy lines are confined. The most eligible options in double-refractory CLL remain non-covalent BTKis, alloHCT, and CAR-T-cells. However, the results of clinical and preclinical trials of several other treatment options, namely BiTEs, PROTACs, novel BCL2is, and CAR-NKs, are promising. Head-to-head evaluations of the aforementioned therapies must be performed to select the superior strategies in the growing formation of double-refractory patients.

## Figures and Tables

**Figure 1 ijms-25-01589-f001:**
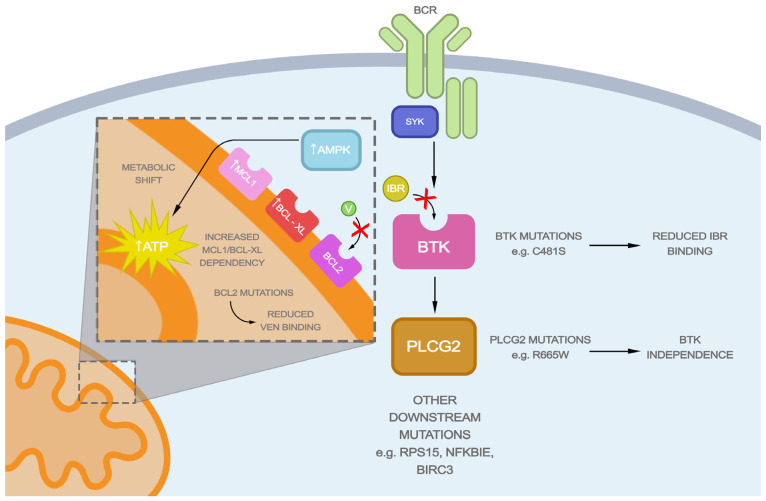
Mechanisms of ibrutinib and venetoclax resistance. In ibrutinib-resistant patients, mutations in BTK or PLCG2 are frequent. The primary BTK mutations, such as C481S, decrease ibrutinib affinity, whereas PLCG2 mutations drive BTK independence. In addition, other downstream mutations (e.g., RPS15, NFKBIE, BIRC3) and chromosomal instability are associated with rendering resistance to ibrutinib. Venetoclax resistance may also be driven by point mutations reducing its binding to BCL2. Additionally, elevated dependency on other anti-apoptotic proteins, such as MCL1 or BCL-XL is seen. Lastly, the resistant cells induce AMPK, which preserves high oxidative phosphorylation, simultaneously decreasing cytochrome c release and thus apoptosis. Abbreviations: AMPK—AMP-activated protein kinase; ATP—adenosine triphosphate; BCL2—B-cell lymphoma two; BCL-XL—B-cell lymphoma extra large; BCR—B-cell receptor; BIRC-3—Baculoviral IAP repeat-containing protein three; BTK—Bruton’s kinase; IBR—ibrutinib; MCL1—myeloid cell leukemia-1; NFKBIE—Nuclear factor of kappa light polypeptide gene enhancer in B-cells inhibitor epsilon; PLCG2—Phospholipase Cγ2; RPS15—ribosomal protein S15; SYK—spleen tyrosine kinase; V—venetoclax.

**Table 1 ijms-25-01589-t001:** Characteristics of BTK inhibitors approved for CLL or in clinical trials.

BTK Inhibitor	Binding and Selectivity	Approval Status	Clinical Indications	Safety	Refs.
Ibrutinib (IMBRUVICA, Janssen, Beerse, Belgium)	Covalent irreversible targeting BTK C481. Inhibits ITK, EGFR, CSK, ErbB2, and TEC	Approved by FDA and EMA	MCL, CLL, WM, MZL, and GVHD	Hypertension, bleeding, atrial fibrillation/atrial flutter	[21,22]
Acalabrutinib (ACP-196, Calquence^®^, AstraZeneca Pharmaceuticals, Cambridge, UK)	Covalent irreversible targeting BTK C481 with high selectivity, reduced off-target effects, no inhibition of EGFR or ITK	Approved by FDA and EMA	MCL, CLL	Atrial fibrillation/atrial flutter (risk lower than with ibrutinib)	[16,22]
Zanubrutinib (Brukinsa, BeiGene, Beijing, China)	Selective, covalent irreversible targeting BTK C481, reduced off-target effects	Approved by FDA and EMA	MZL, CLL, WM	Thrombocytopenia, neutropenia, and bruising	[16,17,22]
Orelabrutinib (ICP-022, HIBRUKA Biogen/Innocare Pharma, Cambridge, MA, USA/Beijing, China)	Covalent irreversible targeting BTK C481 more selectively than ibrutinib	Breakthrough Therapy Designation for RR MCL	MCL	Neutropenia, thrombocytopenia, upper respiratory tracts, and lung infections	[18,23]
Spebrutinib (CC-292, AVL-292, Avila Therapeutics/Celgene, Waltham, MA, USA/Summit, NJ, USA)	Covalent irreversible targeting BTK C481 with high affinity	Phase 1 study in RR CLL/SLL	-	Neutropenia, thrombocytopenia, diarrhea, fatigue, nausea, cough, pyrexia, and headache	[19]
Tirabrutinib (Velexbru^®^, ONO/GS-4059, Ono Pharmaceutical, Gilead Sciences, Osaka, Japan/Foster City, CA, USA)	Covalent irreversible very potent and specific BTKi targeting C481 with greater selectivity than ibrutinib	Phase 1 study in various B-cell malignancies	-	Anemia, neutropenia, thrombocytopenia, pyrexia	[20]
Pirtobrutinib (LOXO-305, Jaypirca, Eli Lilly, Indianapolis, IN, USA)	Non-covalent reversible highly selective, next-generation BTKi, blocks the ATP site of BTK through non-covalent, non-C481-dependent binding	FDA approval, Conditional Marketing Authorization of EMA	MCL, CLL/SLL	Infections, neutropenia, anemia, fatigue, pyrexia	[24,25,26,27]
Nemtabrutinib (MK1026, ARQ 531; ArQule, Inc./Merck Sharp and Dohme, Woburn, MA, USA/Rahway, NY, USA)	Non-covalent reversible highly selective BTKi	Phase 1/2 (NCT03162536)	-	Fatigue, constipation, dysgeusia, cough, nausea	[28]
Vecabrutinib (SNS-062, Viracta Therapeutics, Cardiff, NY, USA)	Non-covalent, reversible highly selective BTKi, no activity on EGFR	Phase 1b/2 (NCT03037645)	-	Fatigue, nausea, diarrhea, thrombocytopenia	[29]
Fenebrutinib (GDC-0853, Roche/Chugai Pharmaceutical, Tokyo, Japan/Basel, Switzerland)	Non-covalent reversible BTKi with strong inhibitory efficacy against a single (C481S) and double (T474S/C481S) BTK variant	Phase 1 study in RR B-cell NHL and CLL (NCT01991184)	-	Fatigue, nausea, diarrhea, thrombocytopenia, headache	[30,31]

Abbreviations: ATP—adenosine triphosphate; BTK—Bruton’s kinase; BTK C481—cysteine-481 of BTK; CLL—chronic lymphocytic leukemia; CSK—C-terminal Src kinase; EGFR—endothelial growth factor receptor; EMA—European Medicines Agency; ErbB2—v-erb-b2 navian erythroblastic leukemia viral oncogene homolog 2; FDA—Food and Drug Administration; GVHD—graft-vs-host disease ITK—interleukin-2-inducible T-cell kinase; MCL—mantle cell lymphoma; MZL—marginal zone lymphoma; NHL—non-Hodgkin lymphoma; RR—relapsed/refractory; SLL—small lymphocytic lymphoma; TEC—Tec protein tyrosine kinase; WM—Waldenström’s macroglobulinemia.

**Table 2 ijms-25-01589-t002:** Clinical trials with novel agents in double-refractory patients with CLL.

Author/Reference		Phase	Previous Treatment	Therapeutic Intervention	ORR/CR	mPFS	Safety
Non-covalent BTKi	Mato A.R. et al. [27]	I/II	Median of three prior therapies; 100% BTKi, 87.9% anti-CD20-Ab, 78.9% chemotherapy, 40.5% BCL2i, 18.2% PI3Ki, 5.7% CAR-T 2.4% allo-SCT	Pirtobrutinib	73.3%/1.6%	19.6 months	Most common AEs of grade ≥3: infections 28.1%, 26.8% neutropenia, 8.8% anemia; discontinuation of therapy due to AEs in 2.6%
	Woyach J. et al. [28]	I/II	Median of four prior therapies; 84% BTKi	Nemtabrutinib	57.9%/2.6%	NR	Most common AEs: 33% fatigue, 31% constipation, 25% dysgeusia, 25% cough, 25% nausea, 25% pyrexia; AEs of grade ≥3 occurred in 68% of participants, discontinuation of therapy due to AEs in 8%
BCL2i	Guièze R. et al. [82]	I	Median of one prior therapy	Sonrotoclax (BGB-11417) +/− zanubrutinib	Monotherapy: 67%/33%; combination therapy: 95%/30%	NR	Most common AEs of grade ≥3 in monotherapy: 50% neutropenia, 25% thrombocytopenia, 12.5% pyrexia; most common AEs of grade ≥3 in combination therapy: 14.1% neutropenia, 1.4% thrombocytopenia, 1.4% diarrhea, 1.4% COVID-19
	Davids M.S. et al. [83]	II	Median of two prior therapies; 12% refractory to BTKi and/or BCL2i	Lisaftoclax (APG-2575) +/− acalabrutinib or rituximab	Monotherapy: 65%/NR; lisaftoclax + acalabrutinib: 98%/NR; lisaftoclax + rituximab: 87%/NR	NR	Most common AEs of grade ≥3 in any group: 26% neutropenia, 12% anemia, 5% thrombocytopenia
	Kwiatek M. et al. [84]	I	Median of three prior therapies; 68% BTKi	LOXO-338	NR	NR	Most common AEs of grade ≥3: 15% anemia, 4% COVID-19
PI3Ki	Brown J.R. et al. [85]	I	Median of five prior therapies; 100% fludarabine, 96% rituximab, 87% alkylating agents	Idelalisib	72%/-	15.8 months	Most common AEs of grade ≥3: 42.6% neutropenia, 20.4% pneumonia, 16.7% thrombocytopenia, 11.1% anemia, 11.1% neutropenic fever
	Flinn I.W. et al. [86]	III	Median of two prior therapies; 93% alkylating agent, 78% monoclonal antibody, 60% purine analog	Duvelisib	73.8%/0.6%	15.7 months	Most common AEs of grade ≥3: 30% neutropenia, 15% diarrhea, 14% pneumonia, 13% anemia, 12% colitis
	Mato A.R. et al. [87]	II	Median of two prior therapies; 86% BTKi, 14% PI3Ki	Umbralisib	44%/4.2%	23.5 months	Most common AEs of grade ≥3: 18% neutropenia, 14% leukocytosis, 12% thrombocytosis, 12% pneumonia, 8% diarrhea
BTK degrader	Tam C et al. [88]	Ia/Ib	NR	BGB-16673	NR	NR	NR
	Linton K. et al. [89]	Ia/Ib	NR	NX-5948	NR	NR	NR
	Mato A.R. et al. [90]	Ia/Ib	NR	NX-2127	33%/NR	NR	Most common AEs of grade ≥3: 35% neutropenia, 15% anemia, 4% hypertension
BiTE	Kater A.P. et al. [91]	Ib/II	Median of four prior therapies; 100% BTKi	Epcoritamab (CD3xCD20 bispecific antibody)	NR	NR	Most common AEs: CRS (100%), fatigue (71%), injection-site reaction (43%), and nausea (43%); no episodes of grade ≥3 CRS were noted
	Patel K. et al. [92]	I	NR	Plamotamab (XmAb13676) (CD3xCD20 bispecific antibody)	NR	NR	Among eight CLL patients there were five AEs of grade ≥3: anemia, thrombocytopenia, neutropenia, lymphopenia, CRS
CAR-T	Turtle C.J. et al. [93]	I/II	Median of five prior therapies; 21% of patients were double-refractory; 100% chemoimmunotherapy, 100% ibrutinib, 25% venetoclax	CD4+ and CD8+ CD19-specific CAR-T cells	74%/21%	8.5 months	Most common AEs: 83% CRS, 33% neurotoxicity; 1 fatal neurotoxicity event
	Siddiqi T. [94]	I/II	Median of five prior therapies; 80% of patients were double-refractory; 100% BTKi, 80% venetoclax, 86% chemoimmunotherapy; 6% SCT, 25% PI3Ki	Lisocabtagene maraleucel (CD4+ CD8+ CAR-T cells)	48%/18%	17.87 months	Most common AEs: 85% CRS, 67% anemia, 62% neutropenia, 50% thrombocytopenia; 5 fatal events, 1 related to treatment due to hemophagocytic lymphohistiocytosis
CAR-NK	Liu E. et al. [95]	I/II	Median of four prior therapies; 18% of patients were double-refractory; 45% BTKi, 18% venetoclax; 36% autoSCT	anti-CD19 CAR-NK	73%/64%	NR	Most common AEs of grade ≥3: 91% neutropenia, 91% lymphopenia, 18% anemia

Abbreviations: Ab—antibody; AE—adverse event; autoSCT—autologous stem cell transplantation; BCL2i—B-cell lymphoma-2 inhibitor; BiTE—bispecific T-cell engagers; BTK—Bruton’s kinase; BTKi—Bruton’s kinase inhibitor; CAR-T—chimeric antigen receptor-positive T cells; CAR-NK—chimeric antigen receptor-positive NK cells; CR—complete remission; CRS—cytokine release syndrome; mPFS—median progression-free survival; NR—not reported; ORR—overall response rate; PI3Ki—phosphoinositide 3-kinase inhibitors; SCT—stem cell transplantation.

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
