# Peer review of "Treatment of Double-Refractory Chronic Lymphocytic Leukemia—An Unmet Clinical Need"

_ijms, 2024, doi:10.3390/ijms25031589_

Round 1
Reviewer 1 Report
Comments and Suggestions for Authors
In the present review, Zygmunciak et al, reviewed the literature on the resistance mechanisms for both ibrutinib and venetoclax in CLL and presented novel approaches for treating the double refractory CLL. The review is well-written, with detailed information of the resistance mechanism of ibrutinib and venetoclax and ongoing clinical studies for double refractory CLL. However, some minor issues should be considered:
1. The percentage for the resistance or refractory to BTKi and BCL2i should be included in the introduction.
2. Elaborate the mechanism of action of BTK inhibitors in CLL.
Comments on the Quality of English LanguageSome grammatical error and typing error need to be improrved.
Reviewer 2 Report
Comments and Suggestions for Authors
This is a comprehensive review that provides an update on a topic of major interest to scientists and clinicians dealing with CLL. A few comments and suggestions for improvement are listed below.
1. The information about pirtobrutinib should be updated with its recent approval for the treatment of adult patients with CLL or SLL who have received at least two lines of therapy, including a BTK and a BCL-2 inhibitor
2. The statement that “the C481S mutation lowers the ibrutinib affinity to BTK” is incorrect. This mutation prevents covalent binding of ibrutinib to BTK, thus converting ibrutinib from an irreversible to a reversible BTK inhibitor.
3. Original references should be provided for the statement “...other mutations including point mutations (RPS15, SF3B1, MGA, BIRC3, NFKBIE, CARD11, XPO1) and chromosomal abnormalities (del8p, del18p, MYC gain/amplification, gain of 2p) may be of importance in rendering resistance to ibrutinib”. The cited reference 35 and 36 are both review articles.
4. The term “del17p mutation” is confusing and should be replaced with “del17p” or “17p deletion”.
5. I don’t believe that the statement “many different indications have been accepted for venetoclax monotherapy or combinations” is correct. As far as I know, venetoclax is currently approved only for CLL, SLL and a subset of patients with AML.
6. I don’t believe that there is sufficient evidence to conclude that “the increased resistance of CLL cells inside lymph nodes due to the increased expression of MCL1, BCL-XL, and BCL2A1 is the consequence of the augmented CD40 and TLR9 signaling interplay”, because BCR signaling occurs primarily in lymph nodes (several papers from Adrian Wiestner’s group) and MCL1 is upregulated by BCR signaling (Petlickovski et al, Blood 2005; Longo et al, Blood 2008).
7. The statement “In the lymph nodes ibrutinib not only decreases the pro-survival stimuli but also decreases the BCR-dependent CCL3 and CCL4, which translates to increased efflux of CLL cells into the blood” is incorrect. The chemokines CCL3 and CCL4 recruit T cells and macrophages, whereas the increased efflux of CLL cells into the blood is believed to be primarily because of inhibition of VLA4 and CXCR4 signaling.
8. The paragraph regarding pirtobrutinib should be rewritten to include the data from the study of Mato et al (N Engl J Med. 2023 Jul 6;389:33-44). I would also suggest to correct the statement “...the progressiveness of the disease during the treatment is not common”, considering that median PFS in this study was only 19.6 months.
Comments on the Quality of English Language
The quality of English Language is good, although it can be improved.
Round 2
Reviewer 2 Report
Comments and Suggestions for Authors
The authors have appropriately addressed my comments
Comments on the Quality of English LanguageThe quality of English Language is good